Dietary supplementation with probiotics regulates gut microbiota structure and function in Nile tilapia exposed to aluminum

Yu Leilei 1 2 3
Qiao Nanzhen 1 2
Li Tianqi 1 2 3
Yu Ruipeng 2
Zhai Qixiao zhaiqixiao@sina.com 1 2 3
Tian Fengwei 1 2 3
Zhao Jianxin 1 2
Zhang Hao zhanghao@jiangnan.edu.cn 1 2 4 5
Chen Wei 1 2 4 6
1 School of Food Science and Technology, Jiangnan University , Wuxi , China
2 State Key Laboratory of Food Science and Technology, Jiangnan University , Wuxi , China
3 International Joint Research Laboratory for Probiotics, Jiangnan University , Wuxi , China
4 National Engineering Research Center for Functional Food, Jiangnan University , Wuxi , China
5 (Yangzhou) Institute of Food Biotechnology, Jiangnan University , Yangzhou , China
6 Beijing Innovation Centre of Food Nutrition and Human Health, Beijing Technology & Business University , Beijing , China
Kormas Konstantinos
Electronic publication date: 2019 Jun 3
Publication date: 2019
Volume: 7
Electronic Location ID: e6963
Received 2018 Nov 6; Accepted 2019 Apr 11
Copyright: ©2019 Yu et al.
Copyright year: 2019
Copyright holder: Yu et al.
License: This is an open access article distributed under the terms of the Creative Commons Attribution License, which permits unrestricted use, distribution, reproduction and adaptation in any medium and for any purpose provided that it is properly attributed. For attribution, the original author(s), title, publication source (PeerJ) and either DOI or URL of the article must be cited.
License URL: https://creativecommons.org/licenses/by/4.0/

Keywords: Probiotic, Lactobacillus plantarum, Gut microbiota, Aquaculture, Nile tilapia, Aluminum

Funding: Natural Science Foundation of Jiangsu Province BK20180603 National Natural Science Foundation of China Key Program 31530056 31772090 31601452 Postdoctoral Science Foundation of China 2018M642166 General Financial Grant from the Jiangsu Postdoctoral Science Foundation 2018K016A Self-determined Research Program of Jiangnan University JUSRP11847 BBSRC Newton Fund Joint Centre Award National First-Class Discipline Program of Food Science and Technology JUFSTR20180102 Collaborative Innovation Center of Food Safety and Quality Control in Jiangsu Province This work was supported by the Natural Science Foundation of Jiangsu Province (BK20180603), the National Natural Science Foundation of China Key Program (No. 31530056, 31772090, 31601452), the Postdoctoral Science Foundation of China (2018M642166), the General Financial Grant from the Jiangsu Postdoctoral Science Foundation (2018K016A), the Self-determined Research Program of Jiangnan University (JUSRP11847), BBSRC Newton Fund Joint Centre Award, the National First-Class Discipline Program of Food Science and Technology (JUFSTR20180102), and the Collaborative Innovation Center of Food Safety and Quality Control in Jiangsu Province. The funders had no role in study design, data collection and analysis, decision to publish, or preparation of the manuscript.

==============================
Backgrounds and aims

Aluminum contamination of water is becoming increasingly serious and threatens the health status of fish. Lactobacillus plantarum CCFM639 was previously shown to be a potential probiotic for alleviation aluminum toxicity in Nile tilapia. Considering the significant role of the gut microbiota on fish health, it seems appropriate to explore the relationships among aluminum exposure, probiotic supplementation, and the gut microbiota in Nile tilapia and to determine whether regulation of the gut microbiota is related to alleviation of aluminum toxicity by a probiotic in Nile tilapia.

Methods and results

The tilapia were assigned into four groups, control, CCFM639 only, aluminum only, and aluminum + CCFM639 groups for an experimental period of 4 weeks. The tilapia in the aluminum only group were grown in water with an aluminum ion concentration of 2.73 mg/L. The final concentration of CCFM639 in the diet was 108 CFU/g. The results show that environmental aluminum exposure reduced the numbers of L. plantarum in tilapia feces and altered the gut microbiota. As the predominant bacterial phyla in the gut, the abundances of Bacteroidetes and Proteobacteria in aluminum-exposed fish were significantly elevated and lowered, respectively. At the genus level, fish exposed to aluminum had a significantly lower abundance of Deefgea, Plesiomonas, and Pseudomonas and a greater abundance of Flavobacterium, Enterovibrio, Porphyromonadaceae uncultured, and Comamonadaceae. When tilapia were exposed to aluminum, the administration of a probiotic promoted aluminum excretion through the feces and led to a decrease in the abundance of Comamonadaceae, Enterovibrio and Porphyromonadaceae. Notably, supplementation with a probiotic only greatly decreased the abundance of Aeromonas and Pseudomonas.

Conclusion

Aluminum exposure altered the diversity of the gut microbiota in Nile tilapia, and probiotic supplementation allowed the recovery of some of the diversity. Therefore, regulation of gut microbiota with a probiotic is a possible mechanism for the alleviation of aluminum toxicity in Nile tilapia.

Introduction

Aluminum, the third commonest chemical element and the most abundant metal on earth, is ubiquitous in the environment. In recent years, environmental aluminum levels have increased due to diverse anthropogenic activities such as water treatment, eutrophic lakes control, mining operations, and industrial landfill (Fernandez-Davila et al., 2012; Garcia-Medina et al., 2011). The aluminum level reaches 5.7 mg/L in certain rivers and lakes in England, the United States, China, and Brazil (Camargo, Fernandes & Martinez, 2009; Da Cruz et al., 2015), research has been shown that concentration of aluminum ranging from 0.1 to 0.2 mg/L can be harmful to fish (Baker & Schofield, 1982; Wang et al., 2013). Excessive aluminum can accumulate in multiple fish tissues and organs and exert adverse effects on the blood circulation and on endocrine, metabolic and reproductive function (Azmat, Javed & Jabeen, 2012). Excess aluminum ions in water were reported to cause mortalities and decreasing population of Atlantic salmon in Norway, southeast Canada, and the northeastern United States (Monette & McCormick, 2008). Aluminum contamination causes economic losses in aquaculture and poses potential human health risks from consumption of aquatic products.

Microorganisms produce a variety of metabolites that can have remarkable effects on the external environment and on the host, including changes in pH, suppression of inflammation, and detoxification (Berdy, 2005; Louis, Hold & Flint, 2014). Hence, the gut microbiota can significantly alter the host’s physiology and its metabolism of nutrients and exogenous toxic substances, and it can shape the microbiome and immune systems (Ni et al., 2014). They may be important mediators of the bioavailability and toxicity of toxic metals. Indeed, long-term toxic metal exposure, including aluminum, lead and chromium, altered the composition of intestinal microbiota (Wu et al., 2017; Zhai et al., 2017).

Lactic acid bacteria probiotics, which are generally derived from humans or food products, are generally recognized as safe (GRAS) strains (Farnworth, 2008) and have been widely applied in various situations, including aquaculture, to improve food safety (Sihag & Sharma, 2012). Some probiotics have been used in the culture of some aquatic organisms to promote growth and control infectious disease. They can improve the host’s intestinal microflora balance and increase the protective effect against pathogenic bacteria (Lahti et al., 2013; Pirarat et al., 2015). For example, L. johnsonii La1, L. rhamnosus LC705, B. lactis Bb12, L. casei Shirota and others can effectively control infection with Vibrio anguillarum, Flavobacterium psychrophilum and Aeromonas salmonicida, to prevent furunculosis in rainbow trout (Nikoskelainen et al., 2001), and L. rhamnosus GG can control infection of tilapia by Edwardsiella tarda and Streptococcus agalactiae (Pirarat et al., 2006; Pirarat et al., 2015).

Our previous study showed that supplementation with probiotic L. plantarum CCFM639, a strain with superb aluminum binding and tolerance abilities, decreased the aluminum level in tissues and alleviated aluminum toxicity by preventing oxidative stress and histopathological changes in tilapia (Yu et al., 2017). However, it remains unclear whether the mechanisms of recovery and alleviation are related to the gut microbiota. Therefore, we investigated alterations in the composition and structure of the intestinal microbiota in tilapia after aluminum exposure and the addition L. plantarum CCFM639 to their daily feed.

Tilapia is one of the most important aquatic species in aquaculture worldwide and is farmed in more than 120 countries and territories (Junning et al., 2018). In 2015, its production accounted for more than 10% of all farmed fish worldwide (FAO, 2015). Moreover, tilapia are recognized as a good biological model because they are easy to handle, culture, and maintain in the laboratory (Korkmaz et al., 2009), and because they display excellent stress sensitivity (Zheng et al., 2016). The aim of the study was to explore whether the regulation of gut microbiota is an aluminum toxicity alleviation mechanism exerted by probiotics in tilapia.

Materials and Methods

Probiotic and fish diet preparation

L. plantarum CCFM639, kindly provided by the in-house Culture Collections of Food Microbiology (CCFM), Jiangnan University (Wuxi, China), was inoculated in MRS broth (Qingdao Hopebio, China) and in a static condition at 37 °C for 18 h. After centrifugation at 8,000 g at 4 °C for 5 min, the medium was removed and the cell pellets were suspended with sterile normal saline solution(0.85%) to one-hundredth of the original medium volume. The pellets were mixed with the fish basal diet using a sterile spreader to distribute the bacterial cells evenly and to achieve a final probiotic concentration of 108 CFU/g in the feed. The formula and nutrient levels of the basal fish diet were consistent with those in previous studies (Yu et al., 2017). The dose of the L. plantarum strain was selected on the basis of previous reports (Heo et al., 2013; Ridha & Azad, 2012; Ridha & Azad, 2016). The bacterial concentration in the fish diet was also confirmed by colony counting. The bacteria-containing feed was prepared weekly and stored at 4 °C before use.

Experimental design

One hundred ninety-two male tilapia were purchased from the Freshwater Fisheries Research Centre of the Chinese Academy of Fishery Sciences in Wuxi and stocked in a cylindrical aquarium (0.6 m2 × 0.85 m) at a fish loading ratio of 1.09 g/L for 3 weeks. The average (±SEM) weight of the tilapia was 34.01 ± 0.19 g. The amount of feed consumed each day was 3% of the average body weight of the fish. The fish were fed manually at 9 am and 5 pm each day. After a 3-week adaptation period, the fish were fasted for 1 day. The four groups are listed in Table 1, including the control group, the probiotic group (639 only), the aluminum exposure group (Al only) and the probiotic intervention group (Al + 639) randomly. The fish in each group were randomly assigned to three tanks with 16 tilapias in each tank. In the aluminum exposure group, the tilapia were grown in water with an aluminum ion concentration (AlCl3•6H2O) of 2.73 mg/L. The selection of this dose was based mainly on the aluminum exposure doses reported in drinking water, rivers and lakes in previous studies (Yu et al., 2017). The test period was 4 weeks, the freshwater was changed and the aluminum level in the water was checked every 2 days. Inductively coupled plasma mass spectrometry (ICP-MS; NexIon-300X; PerkinElmer) was used to determine the amount of aluminum in the water.

Table 1 Experimental groups of tilapia with and without aluminum exposure and probiotic feed.

Group	Experiment time (4 weeks)	
Control	Basic feed + normal water	
639 only	probiotic feed + normal water	
Al only	Basic feed + aluminum water	
Al + 639	probiotic feed + aluminum water	
Notes.

CCFM639 feed, feed containing L. plantarum CCFM639 at a concentration of 108 CFU/g; aluminum water, an aqueous environment containing 2.73 mg/L of aluminum ions.

To ensure that fecal samples were not affected by the high level of waterborne aluminum, they were collected quickly after the water was changed to normal water (without aluminum). After collection of the fecal samples, the aluminum ions were added into the water. At the end of the assay, the tilapia were sacrificed under anesthesia after 24 h of fasting. The entire intestinal tract was removed under aseptic conditions and 0.2 g of intestinal contents was squeezed and collected in sterile tubes for analysis of the intestinal microbiota.

The animal experiments was approved by the Ethics Committee of Jiangnan University, China (JN No. 20151027-1129-3), and all procedures about the care and use of experimental animals followed the guidelines set by the European Community (directive 2010/63/EU).

Determination of fecal aluminum levels

One gram sample of feces was transferred to a microwave digestion tank (OMNI; CEM, UK) with 70% concentrated nitric acid. The microwave digestion system (MARS; CEM, UK) was used. The heating procedure of digestion included three stages: stage 1—power 2000 W, ramp 3:00, temperature 120 °C, hold 3:00; stage 2—power 2000 W, ramp 3:00, temperature 150 °C, hold 10:00; stage 3—power 2000 W, ramp 5:00, temperature 190 °C, hold 16:00. After the temperature fell below 50 °C, the samples were removed and diluted to 50 mL with deionized water. ICP-MS (NexIon-300X; PerkinElmer) was used to determine the amount of aluminum in the samples (Ciavardelli et al., 2012).

RT-qPCR analyses for mRNA expression of L. plantarum in feces

The 0.2 g of fecal samples were collected to extract the total genomic DNA following the instructions of the FastDNA Spin Kit (MP Biomedicals, Santa Ana, CA, USA). The fecal genomic DNA was used as a template, and real-time quantitative polymerase chain reaction (RT-qPCR; CFX96; Bio-Rad, Hercules, CA, USA) was carried out using the specific primer. The primer sequences (5′–3′) referred to previous study (Wang et al., 2018) and as follows, LP-F GGAGCCGCTATTAGTATTTTCAT and LP-R AATACAAGCAAGTCTT185GGACCAG. The Ct value of the fecal sample fluorescence quantification was brought into the corresponding standard curve to calculate the copy number, which was converted into the amount per gram of feces. The standard curve of RT-qPCR was as follows: Ct = −3.1434*lg (copies) + 36.977 (R2 = 0.9913) (Wang et al., 2018).

Analysis of gut microbiota

DNA was extracted from 0.2 g of the contents of the whole intestinal tract using an EZNA DNA Kit (Omega Bio-tek, Georgia, USA) and stored at −20 °C. PCR amplification of the 16S rRNA gene was conducted using the forward primer 515F (5′-barcode-GTGCCAGCMGCCGCGG–3′) and the reverse primer 907R (5′-CCGTCAATTCMTTTRAGTTT–3′) (Zhai et al., 2016). Different samples were distinguished with an 8-base barcode and sequenced using a miseq sequencer (Illumina, Inc., California, USA; illumina miseq PE250). The raw data were quality-filtered and aligned using Trimmomatic and FLASH. The operational taxonomic units (OTUs) were clustered with a 97% similarity cutoff using UPARSE (http://drive5.com/uparse/). The chimeric sequences were identified and removed using UCHIME (http://www.drive5.com/uchime/). The OTU germline type was identified by the RDP classifier (http://rdp.cme.msu.edu/) against the SILVA (SSU115) 16S rRNA database using a confidence threshold of 70% (Gajardo et al., 2016; Huyben et al., 2018).

Data analysis

The experimental results were analyzed and tested using analysis of variance and non-parametric tests. The alpha diversity (Chao and Shannon indices) and the beta diversity (PERMANOVA) of the microbiome were calculated based on the OTU level. Tukey’s post hoc test of one-way analysis of variance (ANOVA) were performed in aluminum and L. plantarum levels of feces. A P value of less than 0.05 was used as a cutoff to indicate a statistically significant difference. The results were plotted using Origin 8.6 software (Originlab, Massachusetts, USA).

Results

Aluminum level in feces

Table 2 (Dataset S1) shows the weekly changes in the fecal aluminum content in tilapia. The fecal aluminum content was very low in the control and CCFM639 only groups, and was markedly elevated after aluminum exposure (P < 0.05). Moreover, the aluminum level in feces also increased as the duration of aluminum exposure increased. CCFM639 treatment promoted the elimination of aluminum in the feces. At the fourth week, the fecal aluminum level in the aluminum only group was 25.67 mg/kg. With the continuous administration of CCFM639, the fecal aluminum levels were significantly greater than those in the Al-only group at each test point (P < 0.05), up to 35.46 mg/kg at the third week.

Table 2 Effects of dietary supplementation with CCFM639 on aluminum contents in Nile tilapia feces.

Group	Aluminum level (mg/kg)	
	0 week	Week 1	Week 2	Week 3	Week 4	
Control	1.13 ± 0.06aA	1.54 ± 0.07aA	1.83 ± 0.14aA	1.80 ± 0.10aA	1.54 ± 0.04aA	
639 only	1.19 ± 0.04aA	1.55 ± 0.05aA	1.68 ± 0.13aA	1.74 ± 0.10aA	1.58 ± 0.06aA	
Al only	1.08 ± 0.13aA	22.33 ± 1.31bB	23.67 ± 1.54bB	25.22 ± 1.32bB	25.67 ± 1.01bB	
Al + 639	1.05 ± 0.08aA	25.67 ± 1.23cB	33.14 ± 2.53cC	35.46 ± 2.05cC	33.79 ± 2.37cC	
Notes.

The data shown are the mean ± SEM for each group. The means with different superscript lowercase letters differ significantly among groups, and the superscript capital letters indicate a significant difference among time-points (P < 0.05).

Quantification of Lactobacillus in feces

Table 3 (Dataset S2) shows that the amount of L. plantarum in tilapia feces in the CCFM639-only group was higher than control group by two orders of magnitude. By week 4, the content of L. plantarum in tilapia feces had been significantly decreased by aluminum exposure, from 105.4 copies per gram of feces to 105.0 copies per gram of feces (P < 0.05), whereas the addition of CCFM639 led to a significant increase in the L. plantarum content in feces to that in the 639-only group.

Table 3 Effects of dietary supplementation with CCFM639 on L. plantarum quantification in Nile tilapia feces.

Group	Number of L. plantarum (log copies/g feces)	
	Week 0	Week 2	Week 4	
Control	5.49 ± 0.19aA	5.47 ± 0.09aA	5.40 ± 0.04aA	
639 only	5.52 ± 0.01aA	7.83 ± 0.12bB	7.73 ± 0.02bB	
Al only	5.49 ± 0.01aA	5.24 ± 0.08aB	4.99 ± 0.05cC	
Al + 639	5.54 ± 0.13aA	7.46 ± 0.01cB	7.31 ± 0.13dB	
Notes.

The data shown are the mean ± SEM for each group. The means with different superscript lowercase letters differ significantly among groups, and the superscript capital letters indicate a significant difference among time-points (P < 0.05).

Intestinal microbial diversity and composition

The numbers of OUTs of each group were 147.3, 172.0, 243.7, 206.3, respectively. Concerning alpha diversity microbiome analysis, Shannon and Chao1 indices were found higher in the 639 only and the Al+639 treatment groups compared to the control (Fig. 1, Dataset S3). As shown in Fig. 2A (Dataset S3), Principal Coordinate Analysis based on Unweighted unifrac distance indicated a overall significant clustering on the fecal microbiota composition (P = 0.002), in which PC1 explained 65.75% of the difference. Moreover, the results of PC1 analysis show that aluminum treatment had the greatest effect on the composition of the gut microbiota in tilapia (Fig. 2B, Dataset S3, P < 0.05).

Figure 1 Alpha diversity results for the gut microbiota of Nile tilapia.

The data shown are the means ± SEM for each group. Asterisks represent significant differences compared to the control group, P = 0.007 and 0.042 for Chao (A), P = 0.006 and 0.041 for Shannon (B).

Figure 2 Principal coordinate score plots for the gut microbiota of Nile tilapia.

(A) Unweighted unifrac distance. (B) PC1 values. The asterisk indicates the statistically significant differences ( P < 0.05) between different groups, ns indicates no statistically significant differences.

Aluminum exposure changed the composition of the gut microbiota in Nile tilapia. The four predominant bacterial phyla Fusobacteria, Proteobacteria, Bacteroidetes, and Firmicutes, accounted for 53.74%, 38.21%, 7.54%, and 0.38%, respectively, in control group (Fig. 3, Dataset S4). Fish exposed to aluminum had a significantly greater abundance of Bacteroidetes and fewer Firmicutes, whereas administration of CCFM639 had the opposite effect. Interestingly, CCFM639 administration led to an increase in the abundance of Fusobacteria and Firmicutes and a decrease in the abundance of Proteobacteria.

Figure 3 Effect of L. plantarum CCFM639 on the relative abundance (relative OTU composition) of the components of gut microbiota in Nile tilapia at the phylum level.

.

As shown in Fig. 4 (Dataset S5), 17 dominant genera were detected, the five most dominant were Cetobacterium, Deefgea, Plesiomonas, Flavobacterium and Cytophagales. Aluminum exposure significantly reduced the abundance of Plesiomonas, Deefgea, and Pseudomonas and drastically increased the abundance of Flavobacterium, Enterovibrio, and the families Porphyromonadaceae and Comamonadaceae (Fig. 5 and Table 4, Dataset S5; P < 0.05). In the tilapia exposed to aluminum, the administration of L. plantarum CCFM639 further led to a decrease in the abundance of Enterovibrio, Comamonadaceae and Porphyromonadaceae (P < 0.05). Unlike the results in the control group, supplementation with L. plantarum CCFM639 greatly reduced the abundance of Aeromonas and Pseudomonas (P < 0.05).

Figure 4 Effects of L. plantarum CCFM639 on the relative abundance (relative OTU composition) of the gut microbiota in Nile tilapia at the genus level.

Figure 5 Relative abundance (relative OTU composition, % ± SEM) of the gut microbiota in Nile tilapia at the genus level.

Relative abundance of Cetobacterium, Deefgea, Plesiomonas and Flavobacterium ; (B) Relative abundance of Cytophagales, Enterovibrio, Aeromonas and Porphyromonadaceae; (C) Relative abundance of Comamonadaceae, Sphaerotilus, Vogesella and Enterobacteriaceae; (D) Relative abundance of Bacillus, Pseudomonas, Duganella, env.OPS_17-norank and Chryseobacterium. Data are expressed as mean ± SEM.

Table 4 Effects of CCFM639 on aluminum-induced changes in relative abundance of Aeromonas, Enterovibrio, Comamonadaceae and Porphyromonadaceae of Nile tilapia.

Group	Aeromonas	Comamonadaceae	Enterovibrio	Porphyromonadaceae	
Control	3.20 ± 0.31a	1.52 ± 1.05a	0.76 ± 0.10a	0.03 ± 0.02a	
639 only	1.31 ± 0.25b	0.45 ± 0.17a	0.62 ± 0.50a	0.09 ± 0.03a	
Al only	3.14 ± 0.34a	5.00 ± 0.82b	1.86 ± 0.08b	0.49 ± 0.06b	
Al + 639	2.57 ± 0.50a	2.46 ± 0.25a	0.87 ± 0.20a	0.04 ± 0.004a	
Notes.

The data shown are the means ± SEM for each group. The different superscript letters represent significant differences between groups (P < 0.05).

Discussion

Aquatic animals, especially fish, have direct contact with the water environment, which contains various pollutants and ever-changing microbiota (Egerton et al., 2018). An excessive aluminum concentration in the aquatic ecosystem can be harmful to the physiological functions of fish, and even threaten their survival (Egerton et al., 2018). In addition, the gut microbiota, which has a close association with fish health, can affect its metabolism, physiology, and immune function, and great inter-individual microbial diversity exists (Sullam et al., 2012). In this study, we adopted a culture-independent technology, next-generation sequencing, which is an up-to-date analytical method that can be widely used for analysis of host microbiota, including that of fish. More specifically, next-generation sequencing can be used to provide detailed information of low abundance microbiota can be provided and the genetic potential of species can even be predicted (Ghanbari, Kneifel & Domig, 2015). Our results show that the dominant four phyla in Nile tilapia were Fusobacteria, Proteobacteria, Bacteroidetes, and Firmicutes. Proteobacteria, Bacteroidetes, and Firmicutes accounted for 90% of the gut microbiota in a previous study of the teleost fishes (Giatsis et al., 2015; Liu et al., 2013; Ran et al., 2015). The intestinal microbiota of fish species including tilapia is characterized by various levels of Actinobacteria, Bacteroidetes, Fusobacteria, Proteobacteria and Firmicutes (Adeoye et al., 2016; Wu et al., 2013). At the genus level, the relative abundance of dominant Cetobacterium (affiliated with Fusobacteria) and Plesiomonas were also consistent with the results of a previous study in zebrafish (Ma et al., 2018).

Despite various studies focusing on aluminum exposure and its effects on various host organs, including the liver, kidneys, and brain (Park et al., 2015), the link between aluminum exposure and the host intestinal microbiota remains unclear. The fish gut microbiota is commonly treated as an organ with a significant role in essential physiological functions and overall health. Consistent with our hypothesis, the fecal aluminum level of tilapia increased significantly when the fish were exposed to aluminum, and CCFM639 supplementation promoted aluminum excretion in feces as its excellent aluminum binding ability (Table 1, Fig. 6). These results are consistent with our previous study in a mouse model (Yu et al., 2016). In addition, the amount of L. plantarum in the tilapia feces increased markedly after the fish ingested feed mixed with a probiotic (Table 2). It is notable that aluminum exposure reduced the numbers of L. plantarum in tilapia feces at week 4 (Table 3), possibly due to a decrease in aluminum-intolerant L. plantarum in the gut caused by aluminum exposure.

Figure 6 Potential protective mechanism of CCFM639 against aluminum induced gut injuries in Nile tilapia.

Environmental exposure to aluminum altered the structure and relative abundance of the intestinal microbiota of Nile tilapia, resulting in a significant decrease in the relative abundance of the phyla Firmicutes and Proteobacteria and the genera Deefgea, Plesiomonas and Pseudomonas and elevated the relative abundance of the phylum Bacteroidetes, and the genera Comamonadaceae and Porphyromonadaceae, Flavobacterium and Enterovibrio (P < 0.05; Figs. 3, 4 and 5). The alpha diversity results show no significant difference between the control and 639 groups, and between the Al only and Al + 639 group. Accroding to previous studies (Qin et al., 2018; Uronis et al., 2011), we hypothesized that administration of a single probiotic CCFM639 may not exert a dramatic effect on the richness of the whole gut microbiota, but it may induce alterations in the abundance of specific genera. A previous study demonstrated that toxic metal exposure can influence the gut microbiota composition in mice (Zhai et al., 2017). Ingestion of aluminum damage the intestinal mucosa and reduce the intestinal barrier function and immune function (Vignal, Desreumaux & Body-Malapel, 2016). Pathogens and opportunistic pathogens in the intestinal tract can take the opportunity to multiply, causing disordered gut microbiota (Berg, 1996). A lower relative abundance of phylum Proteobacteria and a higher relative abundance of genus Cetobacterium were found after aluminum exposure. Exposure to silver led to a similar alteration in male zebrafish (Ma et al., 2018). The pathogen Flavobacterium is considered responsible for several fish diseases, including fry syndrome and bacterial cold water disease, which can cause high mortality levels in young fish (Leal et al., 2010; Nematollahi et al., 2003). The increase in the relative abundance of Flavobacterium may be due to its good aluminum tolerance capacity (more than 2,000 ppm) (Konishi et al., 1994). Similar speculation can be used to explain the increase in the relative abundance of Comamonadaceae. A series of metal-resistant genes and gene clusters was found in the whole-genome sequencing of the Comamonas strain, including arsenic, stibium, copper and so forth (Li et al., 2013; Lin et al., 2015). Wu et al., (2015) studied the virulence of several Enterovibrio and Vibrio strains for zebrafish; the results showed that Enterovibrio had a high level of virulence (LD50 values around 104 CFU/g), and that Vibrio had a moderate level of virulence (LD50 values around 106 CFU/g). Enterovibrio promotes the production of indole, which is a toxin that is harmful to intestinal lactic acid bacteria in excess quantities (Nowak & Libudzisz, 2006; Pascual et al., 2009). Porphyromonadaceae is a family in the of Bacteroidetes phylum that can also exert negative effects on the host (Russell et al., 2015). Therefore, aluminum exposure led to an increase in the relative abundance of some harmful bacteria. The dynamic changes in the gut microbiota can directly affect the intestinal mucosa and indirectly affect the health of the fish (Perez et al., 2010). The results help to further explain the harmful effects of aluminum exposure on the host’s growth and antioxidant system in previous study (Yu et al., 2017).

Probiotics have shown an excellent ability to resist disease and prevent pathogens (Hai, 2015). Probiotic supplementation results in improvements in microvilli density and length, which can increase the absorptive surface of the fish intestines and ultimately enhance the host’s physical barrier against potential pathogens (Standen et al., 2016). The use of probiotics stimulates the proliferation of a few probiotic bacteria and decreases the potential pathogens in fish. Numerous studies have demonstrated the significant effects of probiotics in protecting aquatic animals against infection by pathogens, such as the effects of Bacillus spp. against Streptococcus iniae (Cha et al., 2013), and the effects of Pseudomonas spp. against F. psychrophilum (Korkea-aho et al., 2012). Compared with the Al-only group, the administration of CCFM639 significantly reduced the abundance of Enterovibrio, Comamonadaceae and Porphyromonadaceae, indicating the potential protective effect of CCFM639 treatment against aluminum-induced increases in pathogen. Aeromonas is one of the fish pathogens that causes several fish diseases, including motile aeromonad septicemia, furunculosis, ulcer disease, and carp erythrodermatitis (Miller & Harbottle, 2018). In the 639 only group, the abundance of Aeromonas was decreased, which indicates modification of the bacterial community composition by administration of the probiotic CCFM639. Studies in tilapia have also demonstrated an improvement in gut microbiota with the addition of probiotics (Standen et al., 2015).

Conclusions

In conclusion, the accumulated aluminum in Nile tilapia can be excreted through feces, and oral administration of L. plantarum CCFM639 increased aluminum excretion (Fig. 6). Moreover, environmental exposure to aluminum altered the composition of the gut microbiota, and alteration in the levels of Enterovibrio, Comamonadaceae, and Porphyromonadaceae can be recovered by administration of CCFM639. Therefore, this study provides a further explanation of the protective mechanisms of CCFM639 against aluminum toxicity. Apart from promotion of aluminum discharge through feces, regulation of the gut microbiota with L. plantarum CCFM639 may be an underlying mechanism by which aluminum toxicity is alleviated.

Supplemental Information

Dataset S1 The raw data of Al contents in Nile tilapia feces

Click here for additional data file.

Dataset S2 The raw data of L. plantarum quantification in Nile tilapia feces

Click here for additional data file.

Dataset S3 The raw data of alpha diversity for the gut microbiota of Nile tilapia

Click here for additional data file.

Dataset S4 The raw data of the relative abundance of gut microbiota in Nile tilapia at the phylum level

Click here for additional data file.

Dataset S5 The raw data of the relative abundance of gut microbiota in Nile tilapia at the genus level

Click here for additional data file.

Additional Information and Declarations

Competing Interests

Author Contributions

Animal Ethics

Data Availability

The authors declare there are no competing interests.

Leilei Yu conceived and designed the experiments, performed the experiments, analyzed the data, prepared figures and/or tables, authored or reviewed drafts of the paper, approved the final draft.

Nanzhen Qiao, Tianqi Li and Ruipeng Yu performed the experiments, analyzed the data, prepared figures and/or tables, approved the final draft.

Qixiao Zhai conceived and designed the experiments, prepared figures and/or tables, authored or reviewed drafts of the paper, approved the final draft.

Fengwei Tian and Jianxin Zhao conceived and designed the experiments, contributed reagents/materials/analysis tools, approved the final draft.

Hao Zhang and Wei Chen conceived and designed the experiments, authored or reviewed drafts of the paper, approved the final draft.

The following information was supplied relating to ethical approvals (i.e., approving body and any reference numbers):

The animal experiments was approved by the Ethics Committee of Jiangnan University, China (JN No. 20151027-1129-3), and all procedures for the care and use of experimental animals followed the guidelines set by the European Community (directive 2010/63/EU).

The following information was supplied regarding data availability:

The raw measurements are available in the Supplemental Files.

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
