# Peer review of "Dietary supplementation with probiotics regulates gut microbiota structure and function in Nile tilapia exposed to aluminum"

_PeerJ, doi:10.7717/peerj.6963_

## Round 0.1 · original submission · Major Revisions

Please provide a point-by-point reply to all and each of the reviewers’ comments.

Reviewer 1 ·

Basic reporting

The language is not at a professional level and the text is written in a way not easy to follow. I suggest seeking professional help in order to prepare the manuscript in an acceptable form.
The use of the literature is not sufficient, as much irrelevant information is included.
The figures are acceptable as well as the table, but more information is required in order to be more comprehensive.

Experimental design

The research falls into the journal aims and scope. However, the research question is not very well defined, and the background given is not quite sufficient. The methods described need to be presented in more details.

Validity of the findings

The statistical analysis is not performed properly. More analysis is required in order to be able to reach to the right conclusions. There is a lot of speculation in the discussion, which can be decreased in order to discuss the relevant data.

Additional comments

The authors present a work related to the effects of probiotic dietary supplementation with L.plantarum on the gut microbiota and aluminium toxicity alleviation in the Nile tilapia. The aim of the study is interesting overall, but there are several major concerns that have to be addressed properly in order to reach the right conclusions.
1. The language is not at a professional level and the text is written in a way not easy to follow. I suggest seeking professional help in order to prepare the manuscript in an acceptable form.

2. The abstract should be re-written. The authors should include a brief intro, then state their scientific question and then describe the experimental setup, results and conclusions. It is not easy to follow the way is currently written. Moreover, the authors should not use abbreviations at this stage.

3. The introduction is not informative enough, while it includes irrelevant information. The authors should organise their introduction in a more comprehensive manner in order to:
a. Show why the used the aluminium and how is it relevant to study for fish? Where does Al pollution come from? The reference to Mexico is not relevant to the main question the authors ask.
b. Were there studies before showing that it has an impact on the gut microbiota?
c. Why did they choose to use probiotics and why the specific strain?
d. Why did they choose tilapia as an experimental animal?
e. The rationale of the study should be better explained, and the question better defined.
Moreover, the literature used is not always relevant.

4. In the materials and methods, the description of the experimental setup is insufficient. There are several issues concerning the statistical support in the study and the indices measured in order to reach the conclusions. The widely accepted indices to evaluate changes in the diversity and richness (i.e. Shannon H index and observed OTUs) are not used, and the authors reach to conclusions just by speculating from a descriptive analysis. Therefore, I recommend the authors to use widely accepted measures to evaluate the impact on the microbial diversity and richness. Moreover, when comparing the abundances of specific taxa, parametric methods are not accepted since the normality of the data distribution is not assumed. The authors should use non-parametric tests to evaluate the impact of the different treatments on the abundance of specific taxa, but also for the beta-diversity (principal coordinate analysis, PCoA), Permanova test should be applied. The authors should also state which metric they used for the PCoA.
Several others points that also need to be revised:
Line 88. How did the authors select this concentration and how did they check it in the feeds.
Lines 100-104. How many tanks did they authors use per group? The way is written is quite confusing. How did the authors succeed the Al concentration in the water and what are overall the acceptable limits? Was is in excess in the water?
Lines 105-106. The sentence is not clear. How did the authors collect the faeces in order to make sure they were not affected by the Al concentration in the water?
Line 108. Which part of the intestine was selected? It has been recently shown that different compositions of microbiota can exist between different parts of the Nile tilapia gut and these can be affected by diet or other components, thus it is important to understand that there were no shifts in the microbiome. Where there any intestinal contents after 24h? Did the authors select only the gut digesta or also they sampled the tissue as well?

Lines 115-118. Please refer to the exact method used and add if there is any reference. Moreover, be precise on the exact quantities and methods used.

Line 120. What exact quantity of faeces did the authors use for the DNA extraction?

Line 125. The expression of the copy number per g of faeces can only be used if the authors know the exact amount of faeces used for the DNA extraction.

Table 2. Where did these primers come from? Did the authors design them or originated from another study? Please give more details.

Line 132. Which MiSeq sequencing platform was used?

Line 138 – 141. The authors should perform extra data analysis since the use of ANOVA is not accepted for the microbiome data, non-parametric tests should be applied.

5. The results section needs also support with statistics. I would recommend for Tables 3 and 4, the authors to perform a time-point comparison in order to compare the changes across time. Use different indications for these comparisons within the table (i.e. capital letters instead of small ones).
Section 3.3 needs to be re-written after performing the non-parametric analysis for both the PCoA analysis and for each specific taxa. Moreover, the authors should describe their outcome in the OTU (species)-level for a better resolution. Microbial diversity indices should be also used (i.e. Shannon H index or dominance or richness). Furthermore, there is a major confusion between phyla and genera in the text in lines 187 to 196, this section should be also re-written and better describe the results.

6. The discussion section is too extensive, without proper support based on the data presented. It should be re-written to discuss only the relevant results.
Line 234. How can the authors be sure that the excretion of Al is because of the L.plantarum? Maybe the change in the microbial diversity may have caused this effect.
Lines 235. The authors need a reference to support the study on the mice.
Lines 237-240. This sentence is not clear.
Lines 241-245. The authors have no proper statistics to discuss these results.
Line 247-253. This part is very speculative.
Line 253. Pseudomonas is the genera level. How do the authors know that this is relevant to the probiotic in the previous study? The most probable explanation for the increase in the Flavobacterium seems to be the good Al tolerance. The authors cannot support the other hypothesis with the Pseudomonas.
Lines 256-267. This paragraph is not relevant to the results of the study. Please remove.
Lines 277-278. The authors did not test for diversity analysis. Please first do so.
Lines 278-281. Too speculative.
Lines 281-286. This does not seem to be relevant information with the study.
Lines 300-303. In their previous study, the authors tested the Al-induced tissue damages. Why didn’t the authors check that as well in the present study? It would give a more relevant information on the actual performance of the fish as well as if the changes in the microbial diversity would be indeed beneficial.

7. The figure legends should be more informative. Especially the one presenting the principal coordinate analysis. Which metric did the authors use for this type of analysis?

Reviewer 2 ·

Basic reporting

1) More references and background info are needed in the intro. More information on how aluminium exposed fish affects aquaculture and/or fisheries production and human consumption.
2) I am confused about some word choices in the text. I suggest the authors use the full name for “aluminium” instead of “Al” as Al can be confused with the acronym AI (artificial intelligence). Also, the strain “CCFM639” of the probiotic should only be mentioned once in the abstract and methods since the authors should say “probiotic” instead, in my opinion. Genus names should be in italics font (e.g. Flavobacterium).
3) Several spelling mistakes:
a. Introduction: lite, healthy, metabolism, salmon salar (italics, also include common name: Atlantic salmon), decreased, andmetabolism, microbiota.Therefore,
4) Reference missing from lines 52-54, 66-69 (especially rainbow trout), 71-73.
5) Please define “safe strains” (line 61).

Experimental design

1) In general, more information is needed for future studies to replicate this work. The fish trial and microbiota analysis is sufficient, but the authors need to do more statistical analyses to back up their conclusions. The authors looked at the top 15 bacteria, but did not perform stats on the alpha or beta diversities that are commonly performed for microbiota data.
2) Original primary research is within the Aims and Scope of the journal. Research question is well defined, relevant & meaningful. It is stated how research fills an identified knowledge gap.More info needed on lines 84 (MRS broth company), 86 (percent saline), 87 (basal diet composition and origin), 89 (how did you determine 10^8 CFU/g?), 95 (± SE or SD, please indicate), 96 (fed by hand or feeder?), 101 (freshwater?), 102 (how was the concentration of Al measured? Where was the Al from? How often was this checked?), 108 (how much faeces? Distal or proximal intestine? Squeezed or scrapped? Collected in sterile tubes?), 115 (company of the microwave? what percentage?), 116 (How long is a moment?), 118 (company of the AAS? Reference to the method?), 121 (kit company?), 122 (cDNA kit and qPCR machine and company?), 128 (kit company?), 131 (reference of primers), 132 (company and location of sequencer?), 135 (OUT to OTU), 140 (software company?).
3) Remove “Healthy” (line 92) unless you have scored information of blood and immune parameters that you can include in the manuscript.
4) What is the toxic level of Al for tilapia (lines 102-104)? Is 2.73 mg/L below toxic levels and is >20 mg/L toxic? Please make clear.
5) Remove line 104 as this has been stated already.
6) Table 2 is not necessary and the primers can be stated in the text.
7) Please explain what “improved” means (lines 134 and 135).
8) What database and technique did you use to align the sequences? Please explain. Did you remove OTUs that were mistakenly sequenced and related to chloroplasts and mitochondria (not bacteria)? This is a disadvantage of using short pair-end reads. Please see methods section of Huyben et al. (2018) Applied Microbiol doi:10.1111/jam.13738 and Gajardo et al (2016) Scientific Reports DOI: 10.1038/srep30893.
9) Why were no multivariate statistics performed to look at the alpha and beta diversities (e.g. Shannon diversity and PERMANOVA)? You need to do this to provide evidence that the probiotic resulted in changes to the Al exposed fish. Also, was the data normally distributed? Were the data log transformed to be normalized? Please include this or a normality test, such as Shapiro-Wilk and/or Levene tests. P and F values should be reported as well, or at least in the supplemental table.
10) Table 3 is more of a methodology and should be included in the methods section instead of results.
11) Why is the AL+CCFM639 treatment have 8-10mg/L more Al than the Al only treatment? Could this not be equalised? Please explain.
12) Wrong figure cited (line 237).
13) Reference needed (line 247)

Validity of the findings

1) In general, the conclusions are not clear or well supported and imply that a positive effect and mechanism of the probiotic when this may not exist. Data are not robust as there is a small sample size for microbiota analyses (n=3) and limited amount of stats (no multivariate statistics, only an ANOVA). Most of the statements about changes in bacteria abundance are speculatitive and mostly refer to mammalian models and not fish, thus they authors need to be more careful.
2) The statement that the two groups are significantly different is not supported with statistics. PERMANOVA or MANOVA should be performed to give evidence instead of subjective description of a PCo plot (lines 166-171). What similarity index was used to make the PCo plot? Why are the treatment abbreviations different in the plot?
3) How was diversity reduced (line 176-178)? How did you measure this and did you use stats?
4) How many sequences and OTUs per sample did you find and what was the variation (SD)? How many sequences were lost to unknown, eukaryotes, chloroplasts, etc? Were there any unclassified bacteria as it seems strange that all bacteria found were in the reference database? Please explain.
5) Did you include a negative control of water or buffer in order to determine if the samples had contamination? For example, Duganella was found in high amounts (Fig 3) and has been listed as a common contaminant found in negative controls as according to Salter et al (2014) BMC Biology 2014, 12:87. This possibility should be at least stated in the Discussion.
6) Please add the individual p-values for each OTU (line 189-196).
7) “Components” should be replaced with “OTUs” in Fig 3. Why only 17 OTUs? Is this OTUs with >1% abundance? Please state.
8) Units missing from Fig 4 and same comment as Fig 3.
9) Reword “intimate” (line 213).
10) More references needed (line 225), such as Ran et al. (2015) PLoS ONE 10(12): e0145448. Also, Giatsis et al (2015) Scientific Reports 5:18206.
11) References needed (lines 235, 248, 257, 269, 275).
12) Define “good Al tolerance” (line 255).
13) Define “host” (line 257).
14) The authors use the term “slightly increased the diversity and stability of the tilapia’s gut microbiota” (line 277-278). There is no statistical evidence to support this and stability of microbiota was not investigated as the 16S gut bacteria were only evaluated once at the end of the experiment and not over time. The authors should remove this statement from the discussion and conclusion unless they perform stats that show this. In my opinion the plots and abundance figures seem to show an effect of Al exposure, but no effect of the probiotic. There is natural variation of gut bacteria and a sample size of 3 at one time point is very small, thus a higher sample size or more time points are recommended.
15) Define the “mechanism”. Is it that the probiotic binds Al and then the fish deficate it? Please clarify.
16) The authors say that Al increased negative bacteria and decreased beneficial bacteria (line 292). The authors talk about some pathogenic bacteria, but more discussion of beneficial bacteria is needed. More information about common bacteria found in tilapia is also needed in the discussion.
17) The conclusion talks about the probiotic “alleviating adverse effects” (line 296) but this was not evaluated in this study (eg growth, behaviour, cortisol). The phenomena of the probiotic binding Al and increasing more in the faeces than in the fish is not explained, but it is one of the main conclusions. Figure 5 is only referenced in the conclusion with no clear explanation. If this is what the authors are concluding, they need to analyse Al in the carcass, organs, intestinal lining and/or bones of the fish to see if the Al is depositing more in other tissues to conclude this possible mechanism of reducing Al exposure.

---

## Round 0.2 · Minor Revisions

Please revise your manuscript according to the minor suggestions of one of the reviewers.

Reviewer 1 ·

Basic reporting

The revisions improved the quality of the manuscript

Experimental design

I have several comments concerning the statistical analysis applied addressed below.

Validity of the findings

The authors performed all necessary analysis as requested in order to validate their results.

Additional comments

The authors revisions really improved the manuscript.
However, I do still have several comments concerning the presentation of the results.
Introduction:
Line 61: I would write this as : ‘research has been shown that concentration of aluminium ranging from 0.1 to 0.2 mg/L can be harmful to fish’
Line 64-66: ‘decrease the number of Atlantic salmon’ do the authors mean cause mortalities? Please rephrase this sentence
Line 68-76: The authors are switching the topic to microorganisms – I suggest this part goes in another paragraph
Line 88-94: Move this part concerning the tilapia at the end of the introduction – directly after line 104
So according to the above points the intro should be as following, following line 68:
“Microorganisms produce a variety of metabolites that can have remarkable effects … the composition of intestinal microbiota (Wu et al. 2017; Zhai et al. 2017).
Lactic acid bacteria probiotics, which are generally derived from humans or food products…. control infection of tilapia by Edwardsiella tarda and Streptococcus agalactiae (Pirarat et al. 2006; Pirarat et al. 2015).
Our previous study showed that supplementation with probiotic L. plantarum CCFM639…… is an aluminum toxicity alleviation mechanism exerted by probiotics in tilapia.
Tilapia is one of the most important aquatic species in aquaculture worldwide ……excellent stress sensitivity (Zheng et al. 2016).”

Methods: In the 2.6 Data analysis section, what type of t-test was it applied to compare between treatments?

Results: Section 3.3 Intestinal microbial diversity and composition
please replace ‘In the alpha diversity analyses, the microbiomes of tilapia in 639 only and Al + 639 groups were richer than those in the control group (Figure 1).’ with ‘Concerning alpha diversity microbiome analysis, Shannon and Chao1 indices were found higher in the 639 only and the Al+639 treatment groups compared to the control’.
Please replace: ‘As shown in Figure 2, principal coordinate 1 (PC1) and PC2 were 19.88% and 65.75%, respectively. After treatment with aluminum, the results for the Al-only group and Al + 639 group were mainly concentrated in the upper right quadrant and differed significantly from those of the control group (lower right quadrant).’ With ‘As shown in Figure 2, Principal Coordinate Analysis based on Unweighted unifrac distance indicated a significant clustering between the Al treated versus non-treated groups.’
‘Moreover, the results for the 639 only group were concentrated mainly in the lower left quadrant and also differed significantly from those of the control group.’ Was the separation of 639 with control significant? Did the authors perform a post-hoc Permanova test?
‘The results of principal coordinate analysis show that aluminum treatment and the addition of CCFM639 had the greatest effect on the fecal microbiota composition in tilapia.’ In figure 2 I can see a main separation due to the Al exposure. The authors provide a P-value that shows overall a significant clustering but it doesn’t tell what happens between the treatments. The authors show verify this.

Reviewer 2 ·

Basic reporting

All criticisms have been acknowledged and revised.

Experimental design

All criticisms have been acknowledged and revised.

Validity of the findings

All criticisms have been acknowledged and revised.

---

## Round 0.3 · Minor Revisions

As you can see there is one more thing you will need to revise as they involve statistical robustenesss. I believe it is a minor issue which requires little time to deal with. Please revise and resubmit by depicting to me only these two specific points.

Reviewer 1 ·

Basic reporting

The revisions improved the manuscript, but there is still room for the language improvement. However, I leave this to the editor to decide.

Experimental design

Sufficient

Validity of the findings

Sufficient

Additional comments

The revisions improve the manuscript.
I have minor suggestions:
- Move Line 98-100 'The aim of the study was to explore whether the regulation of gut microbiota is an aluminum toxicity alleviation mechanism exerted by probiotics in tilapia.' as the last sentence of the introduction.
- Line 192: The authors used Tukey t-test. Did they test for normality of their data? A parametric posthoc test needs to fullfil spefic terms. The authors need to verify if this is the case otherwise they need to apply non-parametric analysis.

---

## Round 0.4 · accepted · Accept

Thank you for properly revising the manuscript.